# Methodology for the Accurate Measurement of the Power Dissipated by Braking Rheostats

**DOI:** 10.3390/s20236935

**Published:** 2020-12-04

**Authors:** Domenico Giordano, Davide Signorino, Daniele Gallo, Helko E. van den Brom, Martin Sira

**Affiliations:** 1Istituto Nazionale di Ricerca Metrologica, Strada delle Cacce 91, 10135 Torino, Italy; d.signorino@inrim.it; 2Politecnico di Torino, 10129 Torino, Italy; 3Department of Engineering, Università della Campania “Luigi Vanvitelli”, 81031 Aversa, Italy; daniele.gallo@unicampania.it; 4VSL B.V., Thijsseweg 11, 2629 JA Delft, The Netherlands; hvdbrom@vsl.nl; 5CMI, Okružní 31, 63800 Brno, Czech Republic; msira@cmi.cz

**Keywords:** power measurement, braking rheostat, regenerative braking, current transducer, frequency characterization, chopped current, railway system, DC locomotive

## Abstract

The energy efficiency of transportation is a crucial point for the rail and metro system today. The optimized recovery of the energy provided by the electrical braking can lead to savings of about 10% to 30%. Such figures can be reached by infrastructure measures which allow the recovery of the breaking energy that is not directly consumed by the rail system and dissipated in rheostat resistors. A methodology for the accurate estimate of such energy is valuable for a reliable evaluation of the cost–benefit ratio associated with the infrastructural investment. The energy can be estimated by measuring a braking current flowing in the rheostats. The varying duty-cycle associated with the high dynamic variation, from zero to thousands of amperes, makes the current measurement very challenging. Moreover, the digitization of such waveforms introduces systematic errors that affect the energy estimation. To overcome these issues, this paper proposes a technique to measure the power and energy dissipated by the rheostat of a DC operated train with high accuracy. By means of an accurate model of the electrical braking circuit (chopper and rheostat) and the frequency characterization of the current transducer, a correction coefficient as a function of the duty-cycle is estimated. The method is then applied to data recorded during a measurement campaign performed on-board a 1.5 kV train of Metro de Madrid during normal operation. Using the proposed technique, the estimation of the dissipated braking energy is improved by 20%.

## 1. Introduction

Regenerative braking is widely used since the dawn of electrical railway traction. In the past, all breaking energy produced by the traction motors during the braking stage was dissipated by rheostats placed on-board rolling stock. Nowadays, locomotives have an intelligent system able to direct the braking energy flow towards the overhead contact line if other nearby trains can collect such energy, or towards the braking rheostat to be transformed into heat.

The flat-rate scheme of energy billing, adopted for railway undertakings and used until recently, made the knowledge of the energy dissipated by the braking rheostats marginal. The “new deal” established by the European commission through the Technical Specification on Interoperability [1] imposes billing to be performed on real rolling stock energy consumption. This choice will foster energy saving and thus will transform the railway system in a more sustainable means of transport. In this context, an accurate estimation of the energy dissipated by the braking rheostat will become important information for the determination of the energy costs for railway undertakings. At the same time, this policy pushes railway system operators towards improving their capabilities of re-using the electrical energy produced by the braking.

From a technical point of view, DC railway systems can be considered as closed systems. More precisely, because of the considerable amount of power required by railway systems, DC systems are fed by AC high-voltage grids through diode-rectifiers that allow only unidirectional flow of the energy from the upstream AC system to the railway DC system. Therefore, an imbalance between the absorbed and generated energy generates an increase of the voltage [2,3] and an increase of the energy stored in the stray capacitance of the whole system. This is the only way to store extra energy unless other techniques like batteries, super-capacitors, flywheels, reversible substations [4,5,6], or DC grid connections with domestic loads [7,8,9,10] will be incorporated. As demonstrated by [11], even a proper driving style can reduce such extra energy. Several studies have shown that these massive infrastructural modifications ensure the recovery of the energy dissipated on the braking rheostats from 10% up to 30%. [12,13,14]. Accurate knowledge of this energy plays a key role in many applications, for example in economic analyses. The quantification of the real wasted energy enables to evaluate the potential benefit before an infrastructural investment or to measure its impact afterwards.

Nowadays the electrical braking energy dissipated by the rolling stock is estimated by indirect approaches. A commonly used technique exploits the information of speed and braking effort to compute the mechanical energy provided by the motors operating as generators. Together with the generator efficiency and the duty-cycle imposed to the braking chopper the dissipated braking energy is estimated.

An estimation technique, which has been briefly described in [15], exploits the simultaneous measurement of the instantaneous chopped current flowing through the braking rheostats and the voltage at the input of the breaking chopper. The present paper builds upon the latter method and explores the effect of the distortions introduced by the current sensor, used to measure the chopped current flowing in the braking rheostat, on the estimation of the power dissipated by the braking rheostat. Moreover, this paper discusses and proposes a procedure to improve the simplified measurement model for the power estimation proposed in [15]. Correction factors allowing a more accurate estimation of the dissipated power are defined based on a model involving the braking rheostat, the transducer, and the sampling mechanism. These correction factors are applied to real measurements performed on-board DC locomotives in the framework of the European Project MyRailS [16]. The characterization of a current sensor experiencing chopped waveforms with a high slew rate and a high current variation as in the case of the braking rheostat is a challenging topic [17,18,19,20,21,22]. A methodology for laboratory generation of such waveforms and a reference system to accurately calibrate transducers exposed to such waveforms is described in [23,24], respectively.

The paper is organized as follows: Section 2 describes the measurement campaign that provided the electrical quantities related to the braking rheostats which are the basis of this work. A description of the braking chopper and its simplified model are introduced in the Section 3. A detailed quantification of the systematic error, introduced by the power measurement model, is provided in Section 4. The systematic error introduced by the current sensor and its laboratory characterization are explained in Section 5. Section 6 describes the procedure for the correction of the previously mentioned systematic errors. Section 7 collects all the results related to the systematic errors, their impact on the dissipated power and the application of the proposed correction procedure to a real braking event. Finally, in Section 8 a short summary is presented, the results are discussed and conclusions are presented.

## 2. MyRailS Measurement Campaign

As part of the MyRailS project, a measurement campaign was carried out on-board a train aiming to evaluate the energy consumed by the train, dissipated by the braking resistors and fed back to the overhead line, as well as to estimate the impact of the installation of a reversible station [14] in a particular section of the metro system in Madrid.

Figure 1 schematically shows the input stage of the train with highlighted measurement points. The pantograph voltage *v*_F_(t) is filtered by the low pass filter formed by the inductance *L* and by the bank of capacitors placed in parallel to the traction inverters that control the motors. This filtered voltage *v*_F_(t) is modulated by the chopper and imposed on the braking resistors. During a purely dissipative braking phase, the current through the filter *i*_F_(t) is zero and the output power from the inverters is dissipated in the braking resistors through which the currents *i*_Ra_(t) and *i*_Rb_(t) are measured. The voltage measurements were obtained using two Ultravolt 40TF transducers with full scale range of 40 kV and nominal ratio of 1000:1. The voltage transducers are verified in the laboratory and used in the field for applications from DC to 1 MHz with an accuracy of 0.25%. To measure the total current *i*_P_(t) drawn by the train and the filter current *i_F_*(t), two LEM HOP 2000 current sensors were used. Except for the nominal current, their characteristics are identical to the LEM HOP 800 discussed later. The latter type of sensor is used for the current measurement of auxiliary services, *i*_A_, and the currents through the braking resistors *i*_Ra_(t) and *i*_Rb_(t).

All the signals provided by the presented transducers are digitized and collected thanks to a measurement device, positioned in a control cabinet in the wagon underbody, developed within MyRails project. This device is implemented with the National Instruments (NI) CompactRio 9034 equipped with the NI 9467 module for the synchronization with GPS signal and two NI 9223 acquisition modules (16 bits resolution, 50 kSamples/s, 4 channels with synchronized sampling). All data are stored on SD cards, appropriately replaced before the total occupation of the available space. The acquisition system has been tested for a long time in the laboratory. During these tests, no samples have been lost.

Whereas the other current signals have very slow variations over time, the currents *i*_Ra_(t) and *i*_Rb_(t) flowing through the braking resistors have a dynamic current range larger than 500 A for a chopping frequency of 300 Hz, with a duty-cycle that can vary between 0.5% and 50%.

Particular attention has been paid to the measurement of dissipated energy because of particular metrological difficulties, as described in the following.

## 3. The Braking Chopper: Description and Modeling Approach

During braking, the regenerated energy is sent back to the supply system and recovered if other loads can absorb it. If not, this energy is dissipated on-board by means of rheostats. Sending the energy back to the supply system causes an increase of the voltage at the pantograph. If the voltage increases beyond a predefined limit, the braking chopper modulates the DC voltage across the resistor to dissipate the extra energy [25]. The scheme of the dissipative braking system is shown in the Figure 2a. The duty-cycle of the chopped voltage is proportional to the voltage increase. This leads to a balance between the generation and consumption so avoiding a fast and abnormal increase of the voltage.

The combination of the chopper and braking rheostat can be described as a variable resistance *R*_b_ connected in parallel with the Thevenin equivalent resistance *R*_Th_ representing the supply grid (see Figure 3). If the duty-cycle of the chopper is low, the equivalent resistance *R*_b_ is high; as a consequence, the equivalent parallel resistance is close to *R*_Th_ value; the contribution of the braking rheostat is low. With the increase of the duty-cycle, *R*_b_ reduces and so does the equivalent parallel resistance, such that the voltage of the system reduces when the braking energy is constant.

In our application, the monitoring of the current absorbed by the braking rheostat has been performed on a light train for metro application supplied by a 1.5 kV DC system. The braking rheostat can be easily modeled as a series R-L circuit. The rated value of the resistance *R*_b_ is 3 Ω, whereas the stray inductance *L*_b_ is 36 µH. The model of the dissipative braking system is shown in the Figure 2b. The braking chopper consists of a gate turn-off thyristor (GTO). The fly-back diode allows the discharge of the energy dissipated by the stray inductance *L*_b_ to avoid overvoltage. A rigorous simulation of the GTO is burdensome for the energy analysis. An easy way to simulate the GTO is to substitute it with a pulse voltage generator that switches between the (time-dependent) filtered pantograph voltage of the DC link, *v*_F_(*t*), and zero [17]. When the GTO is open (off), the pulse voltage generator *v*_GTO_(*t*) imposes a voltage equal to *v*_F_(*t*); in this way, according to Kirchhoff’s law, the voltage applied to the braking resistor is zero, and so is the current *i*_R_(*t*). When the GTO is closed (on), the pulse voltage generator imposes a null voltage, as a consequence the voltage *v*_F_(*t*) supplies the braking rheostat. The simulated pulse generator enables to mimic the GTO switch-on and -off time by introducing a rise and fall time. Figure 4 describes the time characteristics of the pulse voltage when the voltage *v*_F_(*t*) is constant and equal to *V*_DC_.

The model of the braking rheostat expressed in the Laplace domain is
(1)$$\frac{I_{R}\left(s\right)}{V_{R}\left(s\right)}=\frac{1}{R_{b}+sL_{b}}$$

It allows to estimate the transfer function that links the chopped voltage to the current flowing through the braking rheostat. An example of the current flowing through the chopper for different chopper duty-cycles δ is provided in Figure 5. The chopped voltage has a peak value of 1.65 kV (the average value experienced on-site). As can be seen for a duty-cycle of *δ* = 0.7%, the current cannot reach the maximum expected value of 550 A because of the delay introduced by the stray inductance of the braking rheostat. For higher duty-cycles the maximum current is reached within approximately 0.1 ms.

It is worthwhile noting that in the specific measurement installation considered (see Figure 1) it is not possible to measure the actual voltage over the braking rheostat *v_R_*(*t*) and instead the value of the filtered pantograph voltage *v*_F_(*t*) was measured.

The evaluation of the average power $\bar{P}$ dissipated by the rheostat is computed over a period *T*. The rheostat gets heated in particular for high duty-cycle and the resistance value can vary by about 15%. There model that uses the product of voltage and current has been preferred to a model that involves the varying resistance value and the current squared [10]. The method used for the estimation of the average dissipated power, is calculated over the chopping period *T*, based on the chopped current as measured by the current sensor $I_{\mathrm{HOP}}$ and voltage at input of the braking chopper$\;v_{F}\left(t\right)$ which is constant and equal to *V*_DC_:(2)$$\bar{P}=\frac{1}{T}\cdot \int_{0}^{T}v_{F}\left(t\right)\cdot i_{R}\left(t\right)\cdot dt=\frac{V_{\mathrm{DC}}}{T}\cdot \int_{0}^{T}i_{R}\left(t\right)\cdot dt$$

This approach introduces two systematic errors. The first is due to the use of the DC voltage at the input of the braking chopper instead of the chopped voltage applied to the rheostat. This results in an overestimation of the dissipated power that depends on the chopper duty-cycle. The second is due the use of current as measured by the HOP transducer. In fact, the step response of the current transducer to the sudden change of current values can remarkably change the measured current values. In the following section, these two systematic effects are analyzed trying to find a method to compensate them.

## 4. Systematic Error Due to Simplified Model of the Measurement

A single voltage pulse of duration *t*_1_ and amplitude *V*_DC_ inside a burst of pulses, is expressed as
(3)$$v_{R}\left(t\right)=\{\begin{array}{c} V_{\mathrm{DC}}\;\;\;\;\;\;\;\mathrm{for}\;0\leq t<t_{1}\\ 0\;\;\;\;\;\;\;\;\;\;\;\;\;\;\mathrm{for}\;t_{1}\leq t<T \end{array}$$
where *T* is the repetition time of the pulses. The braking rheostat is modelled as *R*-*L* (with τ=L/R) series circuit (see Figure 2). The current flowing through the rheostat has the following exponential behavior:(4)$$i_{R}\left(t\right)=\{\begin{array}{c} \frac{V_{\mathrm{DC}}}{R_{b}}\cdot \left(1-e^{-\frac{t}{\tau }}\right)\;\;\;\;\;\;\;\;\;\;\;\;\;\;\mathrm{for}\;0\leq t<t_{1}\\ \frac{V_{\mathrm{DC}}}{R_{b}}\cdot \left(1-e^{-\frac{t_{1}}{\tau }}\right)\cdot e^{-\frac{t-t_{1}}{\tau }}\;\;\;\;\;\;\;\mathrm{for}\;t\geq t_{1} \end{array}$$

It is worthwhile noting that (see Figure 6), in practical applications, *t*_1_ is always less than *T/*2, so the current always reaches zero before the following pulse. The correct mean power, *P*_corr_, associated with the pulse and computed over a time-period *T* is
(5)$${\bar{P}}_{\mathrm{corr}}=\frac{1}{T}\cdot \int_{0}^{T}v_{R}\left(t\right)\cdot i_{R}\left(t\right)\cdot dt=\frac{1}{T}\cdot \left\{\int_{0}^{t_{1}}v_{R}\left(t\right)\cdot i_{R}\left(t\right)\cdot dt+\int_{t_{1}}^{T}v_{R}\left(t\right)\cdot i_{R}\left(t\right)\cdot dt\right\}$$

Since the voltage is zero for *t* > *t*_1_, the second component is zero, and the power can be re-written as
(6)P¯corr=VDC2Rb·{δ−τT(1−e−δTτ)}
where δ is the duty-cycle ($t_{1}=\delta T$). Figure 6 shows the area below the current curve that contributes to the correct power estimation.

Now the question is: what happens if we consider the voltage *v*_F_(*t*) that has a constant behavior rather than the chopped voltage *v_R_*(*t*)? In this case, Equation (4) applies, and the area below the current that contributes to the mean power is shown in Figure 7.

The new mean power that is affected by a systematic error is now:(7)$${\bar{P}}_{\mathrm{wrong}}=\frac{1}{T}\cdot \left\{\int_{0}^{t_{1}}v_{F}\left(t\right)\cdot i_{R}\left(t\right)\cdot dt+\int_{t_{1}}^{T}v_{F}\left(t\right)\cdot i_{R}\left(t\right)\cdot dt\right\}$$

Which is equal to:(8)$${\bar{P}}_{\mathrm{wrong}}={\bar{P}}_{\mathrm{corr}}+\frac{1}{T}\cdot \left\{\int_{t_{1}}^{T}v_{F}\left(t\right)\cdot i_{R}\left(t\right)\cdot dt\right\}$$

Given that the voltage $v_{F}\left(t\right)$ is considered constant and equal to *V*_DC_ and introducing the definition of $i_{R}\left(t\right)$, (4), Equation (8) becomes:(9)$${\bar{P}}_{\mathrm{wrong}}={\bar{P}}_{\mathrm{corr}}+\frac{1}{T}\cdot \frac{V_{\mathrm{DC}}{}^{2}}{R_{b}}\cdot \left(1-e^{-\frac{t_{1}}{\tau }}\right)\left\{\int_{t_{1}}^{T}e^{-\frac{\left(t-t_{1}\right)}{\tau }}\cdot dt\right\}$$

The difference between the wrong power and the correct power is
(10)$$\bar{∆P}=\frac{V_{\mathrm{DC}}{}^{2}}{R_{b}}\cdot \left(1-e^{-\frac{\delta T}{\tau }}\right)\frac{\tau }{T}\left(1-e^{-\frac{T}{\tau }\cdot \left(1-\delta \right)}\right)$$

It is possible to account for this effect by introducing a correction coefficient. Considering that, under real circumstances, the quantity $v_{F}\left(t\right)$ is not constant, the corrected measured values becomes
(11)$${\bar{P}}_{corr}\left(kT\right)=\left[\frac{1}{T}\int_{k\cdot T}^{\left(k+1\right)\cdot T}v_{F}\left(t\right)\cdot i_{R}\left(t\right)\cdot dt\right]\cdot K_{DC}\left(\delta \right);\;with\;k=0,1,2\ldots$$

Table 1 provides the values of the correction coefficient *K_DC_* for different values of duty-cycle, defined as the ratio between ${\bar{P}}_{\mathrm{corr}}$ and ${\bar{P}}_{\mathrm{wrong}}$. The values were obtained for a time period of the chopped signal equal to *T* = 1/300 Hz, a rheostat resistance $R_{b}$ = 3.4 Ω and $L_{b}$ = 36 µH.

## 5. Frequency Characterization of Current Sensors

In the application described in this paper, the current transducer is used under conditions much different from those of common use. For this reason, it is fundamental to know the behavior of the sensor under actual operating conditions; that is, with a pulsed current with amplitude of 500 A and at frequency of 300 Hz [19]. In order to be able to measure such a pulsed current a LEM HOP 800 sensor was selected, which is an open loop Hall effect sensor designed for measurements of DC, AC and pulsed currents. The current range of the sensor is 800 A root-mean-square (RMS), the maximum attenuation in the 10 kHz band is −1 dB, and the nominal voltage isolation is 2 kV. The primary and secondary circuits are galvanically separated. The sensor loop has an openable core, which simplifies the installation on-board the train. Figure 8 shows a schematic representation of the HOP 800 sensor. Table 2 presents the main nominal characteristics of the sensor.

The sensor can be simulated by reconstructing and modeling the frequency response with a transfer function at currents similar to those during a real measurement campaign. The actual frequency response was accurately estimated at the Italian National Metrology Institute (INRiM) using a reference current transducer LEM ITZ Ultrastab, with excellent linearity within 1 to 10 parts in $10^{6}$ (parts per million, ppm). It guarantees an overall accuracy at nominal current, 125 A in this case, at +25 °C better than 12 ppm. Moreover, it has a wide frequency bandwidth from DC up to 500 kHz and it is based on closed-loop fluxgate technology. The output of this transducer is a current, that was measured using a broadband 1 A, 100 kHz 7320 Guildline shunt. The frequency dependence of the shunt varies from essentially zero at power frequencies to less than 10 ppm up to 10 kHz. For the current generation a Clarke-Hess Model 8100 trans-conductance amplifier has been used. The six overlapping ranges of the amplifier, 2 mA, 20 mA, 0.2 A, 2 A, 20 A, and 100 A, provide output currents with distortion less than −60 dB up to 10 kHz (typically 20 kHz) and less than −40 dB to 100 kHz at all current ranges. The input signal for the trans-conductance amplifier was generated using a National Instruments 5422 digital-to-analogue converter. The NI 5422 is a waveform generator with the following features: sampling frequency up to 200 MSa/s, 16-bit resolution output channel, full scale range 12 V (peak-to-peak) for a 50 Ω load up to 80 MHz, DC offset up to ± full scale and a bandwidth of 80 MHz [26].

A National Instruments PXIe-6124 was synchronized to the generator and used to measure the voltage to characterize the HOP sensor. The NI 6124 is a multi-functional simultaneous sampling data-acquisition device, with a 16-bit analog-to-digital converter and a sampling rate up to 4 MSa/s per channel. Figure 9 shows the block diagram of the setup used to characterize the current sensor. For specific frequencies in the range from 47 Hz to 40 kHz the gain has been evaluated 30 times with an averaging time of 1 s.

The gain for the i-th frequency, $G_{\mathrm{dB}}\left(f_{i}\right)$, expressed in dB, can now be defined as follows:(12)$$G_{\mathrm{dB}}\left(f_{i}\right)=20\log_{10}\left(\frac{I_{\mathrm{HOP}}\left(f_{i}\right)}{I_{\mathrm{ITZ}}\left(f_{i}\right)}\right)=20\log_{10}\left(\frac{K_{\mathrm{HOP}\_\mathrm{DC}}*K_{\mathrm{ch}1}*V_{\mathrm{HOP}}\left(f_{i}\right)}{K_{\mathrm{ITZ}}*K_{\mathrm{GUI}}*K_{\mathrm{ch}0}*V_{\mathrm{REF}}\left(f_{i}\right)}\right)$$
where $I_{\mathrm{HOP}}\left(f_{i}\right)$ and $I_{\mathrm{ITZ}}\left(f_{i}\right)$ are the magnitude of the phasors of the primary current $I_{\mathrm{pr}}$, at frequency $f_{i}$, measured by the HOP sensor and the ITZ sensor respectively. The current $I_{\mathrm{HOP}}$ can be written as the product of the voltage measured by the acquisition system, $V_{\mathrm{HOP}}\left(f_{i}\right)$, multiplied the channel gain, $K_{\mathrm{ch}1}$, and the gain of the HOP transducer in stationary conditions, $K_{\mathrm{HOP}\_\mathrm{DC}}$. In the same way, the current measured by the reference transducer can be written as the product of the channel gain,$\;K_{\mathrm{ch}0}$, the reference shunt gain,$\;K_{\mathrm{GUI}}$, and the gain of the reference transducer,$\;K_{\mathrm{ITZ}}$, multiplied by the voltage read by the acquisition system,$\;V_{\mathrm{REF}}\left(f_{i}\right)$.

Although the transconductance amplifier has the capability to reproduce signals up to 500 kHz, the generation of the current is limited by the circuit connected to it. With an inductive load, increasing the frequency corresponds to an increase in the impedance and therefore an increase in the voltage required by the amplifier for generating the output current. The compliance voltage for this amplifier is 7 V. In order to minimize the impedance, the first tests were carried out with twisted cables as shown in the Figure 10a.

The result of this characterization shows that, in this configuration, the accuracy of the instrument is highly out of specification. This effect is due to the position of the Hall sensing elements in the current transducer which are influenced by the magnetic field produced by the cables in the vicinity. Subsequently, by reducing the maximum test frequency, tests were carried out with non-twisted cables, as shown in Figure 10b. The effect of the position of the primary conductor on the transducer scale factor has been investigate and described in [27]. This activity was limited to the position of the primary conductor rather than the effects of a current flowing close to the hall sensor. Moreover, a limited frequency band was analyzed.

Once the frequency characterization has been obtained, it is possible to proceed with the search for the poles and zeros that define the transfer function of the current sensor. By following the approach already presented in [28], the transfer function of the current sensor was fitted by the relation between the output current *I*_HOP_ and the input current *I*_p_ that best approximates the actual frequency behavior of the scale factor in the Laplace domain:(13)$$\frac{I_{\mathrm{HOP}}\left(s\right)}{I_{p}\left(s\right)}=\frac{\left(s-\omega_{01}\right)\cdot \left(s-\omega_{02}\right)}{\left(s-\omega_{p1}\right)\cdot \left(s-\omega_{p2}\right)}$$

The function is constituted by two zeros and two poles. The model allows for the estimation of the correction factors as a function of the duty-cycle that were applied to the power and energy estimated by the on-board locomotive measurements, in order to compensate for the systematic errors.

Further, in this case it is possible to account for systematic errors by introducing a correction coefficient. The correction coefficient representing the sensor distortion and the limited sampling frequency can be defined as the ratio between the integral of the measured current squared and the current flowing through the rheostat squared:(14)$$K_{\mathrm{HOP}}=\frac{\int_{0}^{T}i_{p}^{2}\left(t\right)\cdot dt}{\int_{0}^{T}i_{{\mathrm{HOP}}_{Z}}^{2}\left(t\right)\cdot dt}$$

The integrals are numerically computed by implementing the trapezoid rule.

## 6. The Methodology for Accurate Estimation of the Dissipated Power

As shown in the previous section, the systematic deviations due to the voltage-and-current measurement method can be accounted for by correction factors. So, the full measurement model now becomes
(15)$${\bar{P}}_{\mathrm{corr}}\left(kT\right)=\left[\frac{1}{T}\int_{k\cdot T}^{\left(k+1\right)\cdot T}v_{F}\left(t\right)\cdot i_{R}\left(t\right)\cdot dt\right]\cdot K_{\mathrm{DC}}\left(\delta \right)\cdot K_{\mathrm{HOP}}\left(\delta \right);\;\mathrm{with}\;k=0,1,2,\ldots$$

As shown in the previous section, the error is dependent on the value of the chopper duty-cycle *δ*. However, there is no direct information on duty-cycle because the chopped voltage is not measured. So, the duty-cycle should be estimated using the current measurements, but there is a difference between the duty-cycle measured on current and measured on voltage because of the distortions introduced by the stray inductance of the rheostat and by the current sensor. In particular, the duty-cycle of the chopped current is always higher than the chopped voltage duty-cycle (see Figure 5). Thanks to the model of the braking rheostat that provides the current behavior by applying a pulsed voltage of known duty-cycle, the relation between the duty-cycle of the chopped current, *δ*_I,_ and the duty-cycle of the chopped voltage, *δ*_V,_ applied to the rheostat can be identified. The relation between *δ*_V_ and *δ*_I_ is shown in Figure 11 for the lowest values.

This relation is fitted using the ad hoc function:(16)$$\delta_{V}=a\cdot sin\left(\delta_{I}-\pi \right)-b\cdot {\left(\delta_{I}-10\right)}^{2}+c$$
where *a*, *b,* and *c* are equal to 4.253, 0.2636 and 26.36, respectively. Formula (16) has been carried out by the black-box approach, exploiting the optimization algorithms provided by Matlab software.

The procedure that provides the corrected power dissipated by the braking rheostat is summarized in Figure 12. The measured chopped current is processed in order to determine the duty-cycle *δ*_I_ for a specific period *T* and to calculate the power dissipated by the rheostat *P*. From this, it follows that the minimum time for the estimation of the duty-cycle and the relative correction of the dissipated power is a single period of the chopped signal, in this case 3.34 ms. In order to reduce the impact of a single outlier on the estimated power, such time-interval can be enlarged of an integer number of times.

## 7. Results

This section provides the results of the sensor characterization in the frequency domain, i.e., the fit function that approximates the frequency dependence. A comparison between the actual current flowing through the rheostat and the distortion introduced by the current sensor and the subsequent digitization is provided. Furthermore, the behavior of the correction factors as a function of the duty-cycle are described, including a description of the uncertainty introduced by the initial phase of the sampling procedure that is considered as stochastic information. Finally, the impact of the corrections of the estimation on real power and energy dissipated by a braking rheostat on an Italian DC locomotive for commuter service is shown.

### 7.1. Frequency Behavior of the Current Sensor

The results of the measurement campaign regarding the frequency dependence of the current sensor gain, and the subsequent identification of the function that describes the frequency characteristics with sufficient reliability, are presented in the next subsections.

#### 7.1.1. Measurement Results of the Frequency Dependence of the Correction Coefficients

The two arrangements of the primary conductor described in Section 5 provide considerably different frequency behavior, in particular around 10 kHz [29].

As can be seen by Figure 13 the effect of different primary cable arrangement is very high. The frequency behavior is compliant with the datasheet of such commercial sensor if the forward and return cables of the primary circuit of the sensor are kept far from the yoke. In fact, for case (b) the gain is about −1 dB at 10 kHz, as declared in the datasheet. The frequency band in the twisted cable configuration is considerably lower than rated, i.e., about −1 dB at 2 kHz.

#### 7.1.2. Fitting the Dependency of the Correction Coefficient on the Frequency

The function that approximates the frequency behavior of the current sensor has been tailored considering the behavior obtained with the cables far from the yoke of the magnetic circuit of the sensor (red curve in Figure 13). It can be deduced that an overshoot occurs around 5 kHz. Such behavior can be reproduced by a couple of two complex and conjugate poles. Moreover, one can note that the slope of the measured frequency dependence is lower than the expected 40 dB per decade produced by the two poles; this is taken into account by introducing a zero at about 3 kHz that reduces the slope and allows a better fit in the frequency range between a few kilohertz to 10 kHz. The description of the fit function *G*_fit_, in the Laplace domain, is based on Equation (13) described as follows:(17)$$G_{\mathrm{fit}}=\frac{\left(s-\omega_{01}\right)\cdot \left(s-\omega_{02}\right)}{\left(s-\omega_{p1}\right)\cdot \left(s-\omega_{p2}\right)}$$
(18)$$\begin{array}{c} \omega_{01}=-2\pi \cdot 4.3\cdot 10^{3}\\ \omega_{02}=-2\pi \cdot 20\cdot 10^{3}\\ \omega_{p1},\;\omega_{p2}=-\sigma \cdot \omega_{p}\pm j\omega_{p}\cdot \sqrt{1-\sigma^{2}}\\ \sigma =0.91,\;\omega_{p}=2\pi \cdot 6.3\cdot 10^{3} \end{array}$$

A comparison between the fit functions and the measured frequency dependence of the sensor gain is provided in Figure 14. The fit agrees with the measurement data within 0.5 dB up to 10 kHz.

### 7.2. The Actual Current Flowing in the Braking Rheostat

Using the transfer function of the braking rheostat and the current sensor, it is possible to compare the time behavior of the simulated-actual current, $i_{\mathrm{sim}}$, flowing in the braking rheostat, the distorted signal provided by the current sensor, $i_{\mathrm{HOP}}$, its digitization, $i_{{\mathrm{HOP}}_{Z}}$, and the samples acquired on-board the locomotive, $i_{\mathrm{measured}}$. Figure 15 provides a comparison among such quantities for two different vales of the duty-cycle *δ* of the braking chopper: *δ* = 0.9% that is the minimum recorded value (Figure 15a) and *δ* = 15 % (Figure 15b).

For low duty-cycle, the distortion introduced by the current sensor is much more evident, the peak value of the actual current is not detected. For higher duty-cycle, the overshoot on the signal provided by the sensor, shown in Figure 15b, is the effect of the two complex poles. The good overlap between the measured current and the signal provided by the simulation demonstrates the accuracy of the frequency characterization of the current transducer.

### 7.3. Sampling Effects in the Dissipated Power/Energy Estimation

The sampling mechanism, in particular for pulses with a low duty-cycle, can dramatically affect the area underneath the sampled curves. Moreover, the position of the first sample following the onset of the pulsed signal can provide different results in terms of the area estimation and thus provide different power estimation. A demonstration of the size of this effect is shown in Figure 16a for a duty-cycle of *δ* = 0.9%. The effect in the time domain of a different position of the first sample is provided in Figure 16b. In particular, the figure shows the position of the samples that provide the lowest and biggest deviation between the area underneath the actual current wave and the sampled signals.

### 7.4. Estimation of the Correction Coefficient as a Aunction of the Current Duty-Cycle

The correction coefficient *K*_HOP_, defined in Section 5, strongly depends on the duty-cycle of the chopped voltage signal. In detail, *K*_HOP_ is high with a low duty-cycle, where the distortion effect introduced by the current transducer is higher. Such effect is also evident in Figure 17. As described in the previous subsection, the value of *K*_HOP_ depends on the position of the samples in time with respect to the position of the pulse. This information is not deterministic but random; as a consequence, for each duty-cycle a distribution of possible values of *K*_HOP_ is found by varying the temporal position of the samples. This procedure provides a bandwidth of possible values of *K*_HOP_ for each duty-cycle value. Figure 17a describes the behavior of *K*_HOP_ as a function of the duty-cycle and the error bars indicate the bandwidth of possible values.

The correction coefficient *K*_DC_ that compensates for the error in the definition of the power dissipated by the rheostat is significant for low duty-cycle and tends to unity with increasing duty-cycle, as shown in Figure 17b. Some values of the two correction coefficients and their product with the associated interval of maximum variability due to the position of the samples in time with respect to the pulse onset position is provided in Table 3.

### 7.5. Impact of the Offset Introduced by the Current Sensor

During a real measurement, the acquisition system shows an offset caused by external influence factors or internal offsets of the sampling device. This offset manifests itself even when the chopped voltage is switched off and the power is zero. If neglected, the offset can introduce an error of up to 10% of the whole braking energy during the campaign. Since the offset level is changing in time, it has to be determined for every braking pulse independently. The offset level is estimated using the samples before and after a braking pulse and subtracted from the braking pulse.

### 7.6. Application of the Correction Methodology to Measured Data

The proposed procedure is applied to a single electrical braking event recorded during a commercial route of a train operating on line 10 B of Metro de Madrid [30]. The braking event lasts for about 10 s. The DC voltage at the input of the chopper is 1780 V on average (see Figure 18a). The time behavior of the two input parameters of the methodology proposed for the correction, *δ*_I_ and the power *P* have been calculated by means of the formula shown in Figure 12, over a time window of 20 ms (six periods of the chopped signal). Such integration time allows to reduce the impact of a single outlier on the estimated quantities without losing the information on their dynamic variations. The variation of these quantities with time is shown in Figure 18b. The dissipated power is higher at the beginning of the braking, increasing to a maximum of 160 kW; after a couple of seconds the power decreases to about 40 kW. Two power peaks of 60 kW and 80 kW are reached at about 6.5 s and 8.5 s.

As can be seen from Figure 18b, the power behavior with time follows the behavior of the duty-cycle *δ*_I_, computed from the chopped current. The maximum value of *δ*_I_ of about 15% is reached in the first instant of the braking event, whereas the minimum is about 1.8%.

The time behavior of *δ*_I_ and the analytical function (16) allow the estimation of the duty-cycle of the chopped voltage and, as a consequence, the time behavior of the correction coefficients *K*_DC_ and *K*_HOP_, which is shown in Figure 19a. As can be seen, the weight of the combined correction coefficients (*K*_DC_∙*K*_HOP_) ranges from about 0.95 to 0.75. The value of *K*_HOP_ provided here refers to the mean value. A comparison between the power before the correction, *P*, and the values obtained after the application of the correction *P*_corr_, is reported in Figure 19b. The same figure provides the behavior of the percentual deviation Δ between the two powers computed as
(19)$$\Delta \;=\frac{P-P_{\mathrm{corr}}}{P_{\mathrm{corr}}}\cdot 100$$

The deviation between non-corrected and corrected power ranges from 5% to about 34%; as expected, the deviation is higher for lower power. The considered braking is characterized by an average power dissipation, for a single rheostat, of about 35.8 kW.

As described in Section 7.4, the correction factor *K*_HOP_ is characterized by a range of possible values for each duty-cycle value (Figure 17a). Such random behavior propagates into the power estimation. The impact of the correction factor on the power is about 11% with an associated range of values of ± 2%.

To give evidence of the impact of the correction on the measured power, a collection of values for the two power magnitudes and their deviation is provided in Table 4. The table also provides the interval of variability (±var) associated with the corrected power *P*_corr_ and, as a consequence, to the deviation due to the variability on *K*_HOP_.

## 8. Discussion and Conclusions

A methodology for the accurate estimation of the power and energy dissipated by a braking rheostat installed on-board rolling stock has been presented. A description of the measurement procedure applied on-board a metro vehicle and a model used for the power estimation is provided. The treatment focuses on the description, modeling and quantification of two systematic errors that affect the power estimation. One is due to the approximate model for the power estimation, a consequence of practical implementation of the measurement setup that has to guarantee an adequate level of safety, reliability and has to reduce, as much as possible the invasiveness towards the electrical apparatuses of the locomotive. The other error is introduced by the limited bandwidth of the current sensor employed in the chopped-current measurement and the limited sample frequency for the digitization of the signal. The variability in the power estimation introduced by the limited sample frequency is studied and quantified. It has been demonstrated that the correction factors dramatically affect the power estimation for low power amplitude, that is, low duty-cycle. For a duty-cycle of about 1%, which is the minimum duty-cycle experienced, the corrected power is reduced by about 36%. The same occurs for the range of variability introduced by the stochastic positions of the sampled data with respect to the current pulse. The variability at 1% duty-cycle ranges from 17% to 41% with an average value of 31%. This means that, as expected, the uncertainty associated with the power estimation is higher for low power amplitude. We conclude that long braking with low duty-cycle, that is, low dissipated power, is estimated with a high uncertainty.

The proposed technique has been applied to a real dissipative braking event recorded on-board a metro vehicle. This braking event, during about 10 s, is characterized by a duty-cycle ranging from 5% to 14%; with the same time behavior, the power ranges from few tens of kilowatts to about 130 kW with an average value of about 60 kW. Under these conditions, the deviation between uncorrected and corrected power is about 20%. Such an error is relevant for proper energy flow analysis as in the design of on-board storage systems. This demonstrates the usefulness of the proposed methodology.

## Figures and Tables

**Figure 1 sensors-20-06935-f001:**
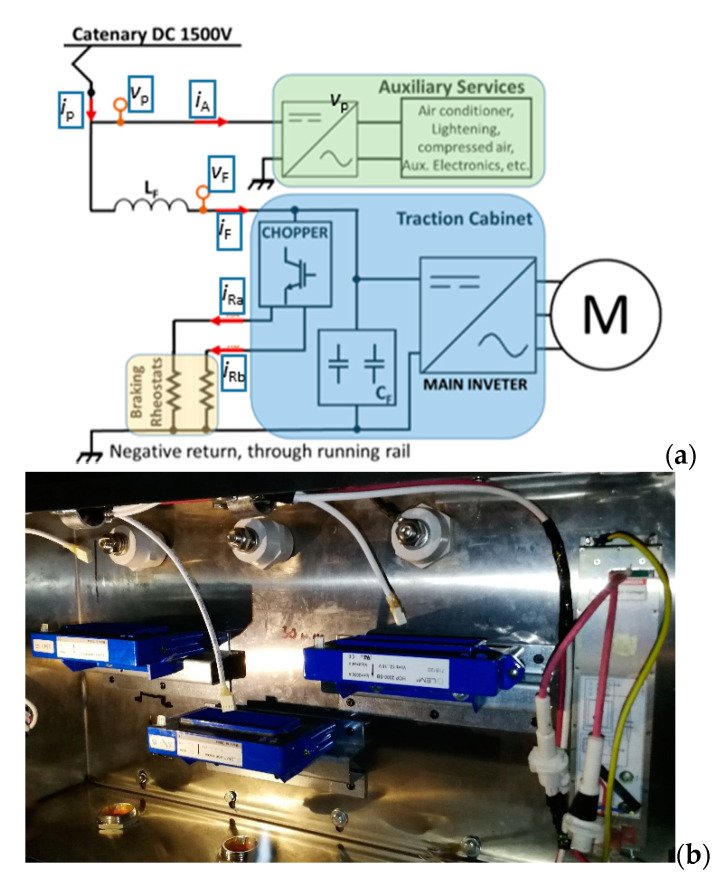
Scheme of the input stage of the metropolitan electro-train and identification of the electrical quantities monitored during the measurement campaign (**a**). Arrangement of the current sensors measuring the quantities *i*_F_, *i*_Ra_ and *i*_Rb_ (**b**).

**Figure 2 sensors-20-06935-f002:**
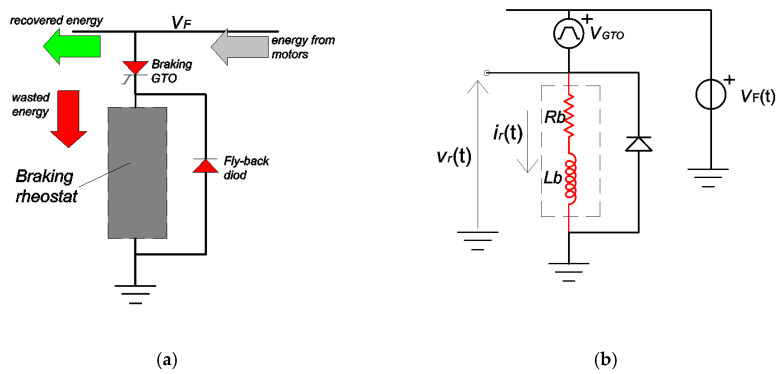
(**a**) Circuit modeling the electrical braking; (**b**) Same circuit though with a simplified model of the braking chopper (GTO).

**Figure 3 sensors-20-06935-f003:**
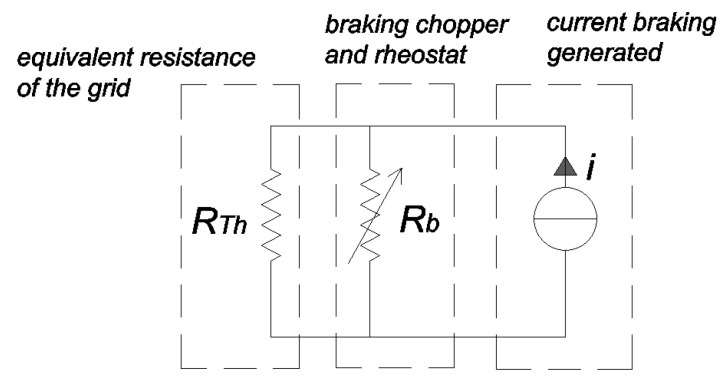
Circuit describing the effect of the braking rheostat.

**Figure 4 sensors-20-06935-f004:**
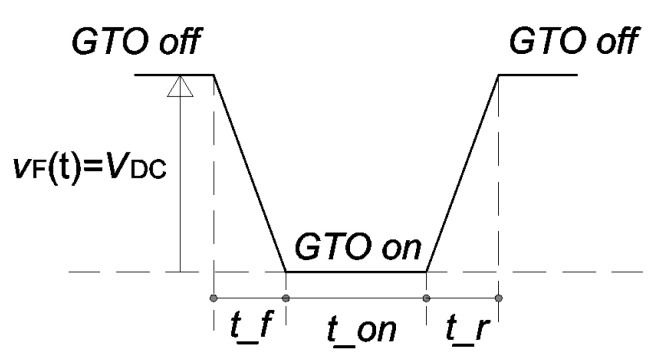
Time characteristics of the voltage pulse generator simulating the voltage across the rheostat resistor.

**Figure 5 sensors-20-06935-f005:**
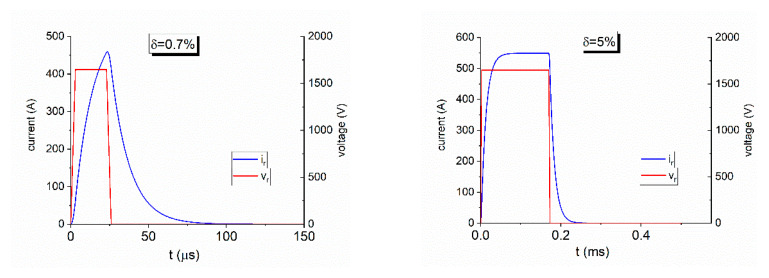
Chopped voltage (red) and current (blue) in the braking rheostat for different duty-cycle values.

**Figure 6 sensors-20-06935-f006:**
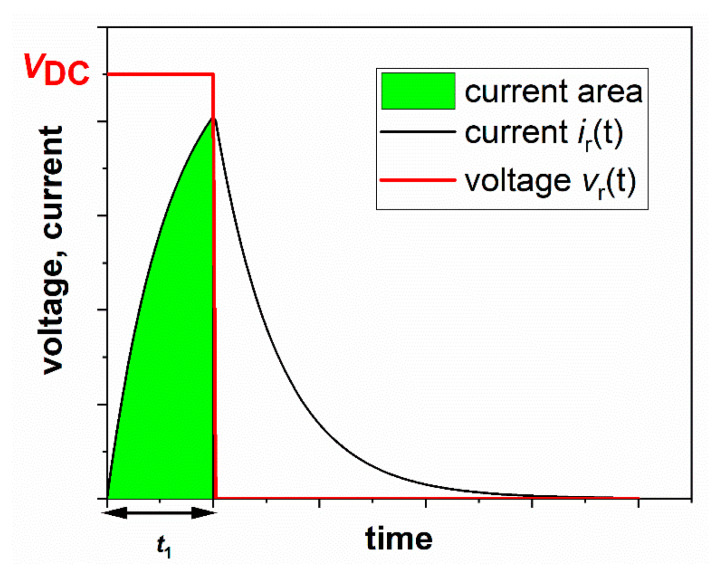
Area below the current curve that correctly contributes to the mean power estimation.

**Figure 7 sensors-20-06935-f007:**
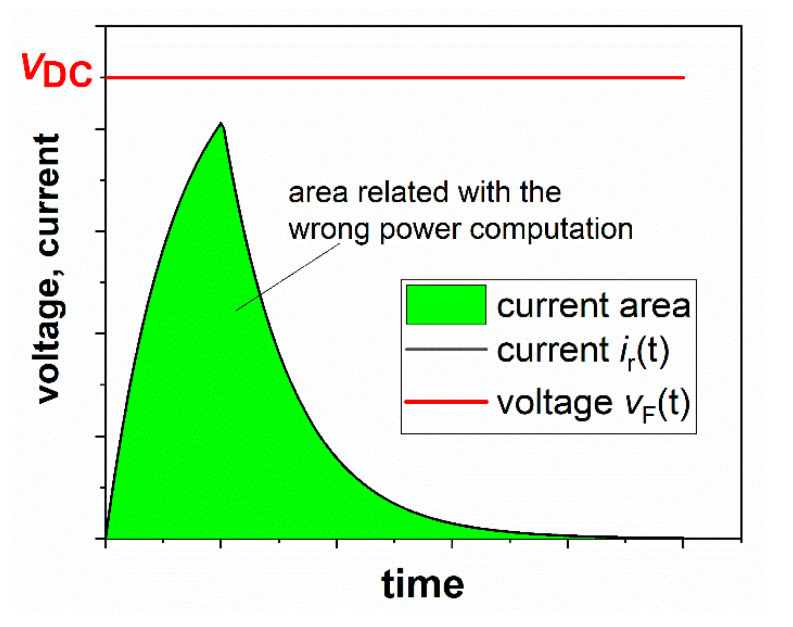
Area below the current wave that contributes to the wrong estimation of the mean power.

**Figure 8 sensors-20-06935-f008:**
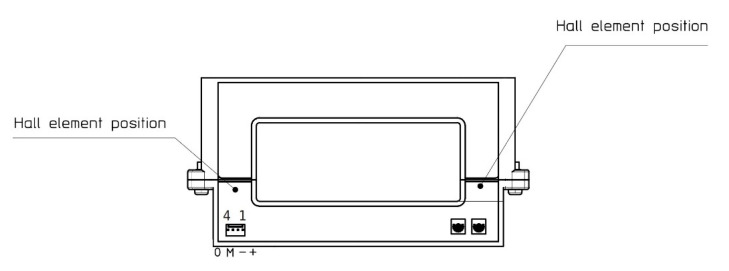
Schematic representation of the HOP current sensor.

**Figure 9 sensors-20-06935-f009:**
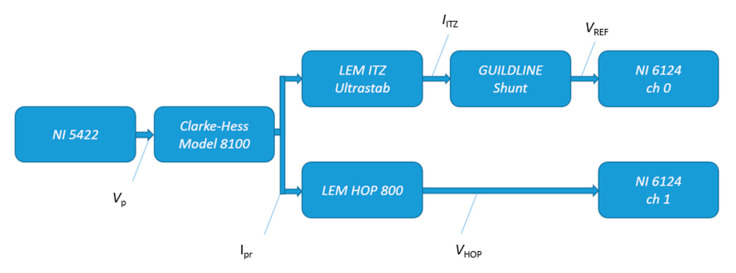
Scheme of the set-up for the frequency characterization of the current sensor.

**Figure 10 sensors-20-06935-f010:**
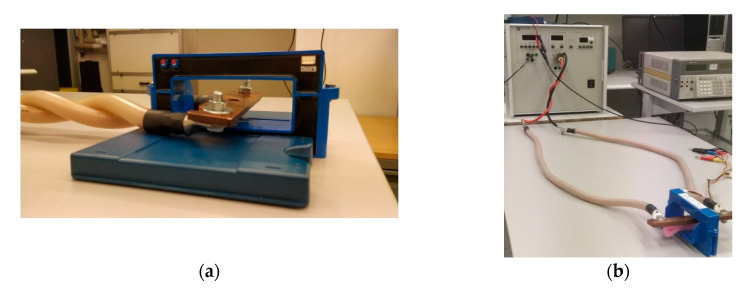
Pictures describing the two different primary cable arrangements. Twisted cables close to the magnetic yoke of the HOP sensor (**a**); supply cable far from the magnetic yoke of the HOP sensor (**b**).

**Figure 11 sensors-20-06935-f011:**
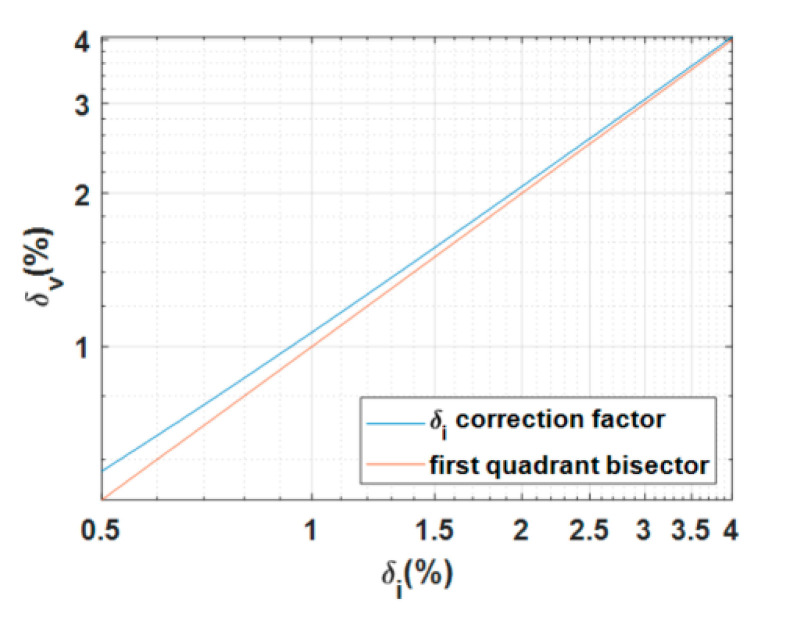
Duty-cycle *δ*_V_ versus *δ*_I_. The first quadrant bisector is shown for indicative purposes only.

**Figure 12 sensors-20-06935-f012:**
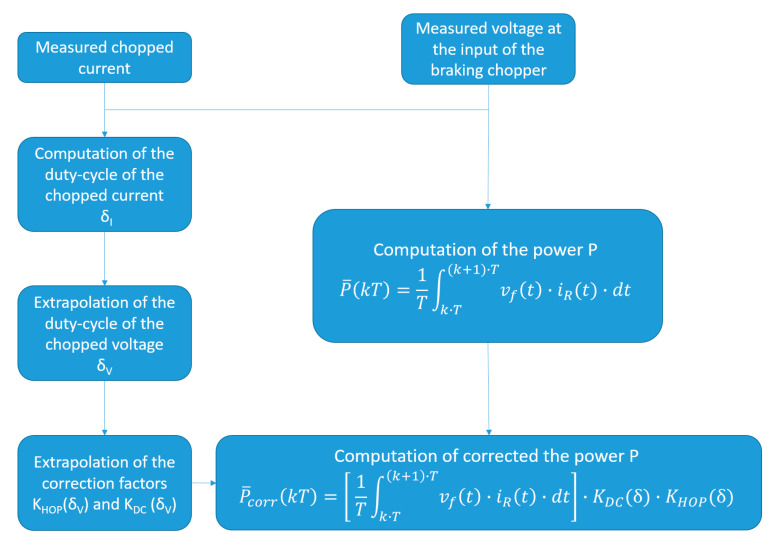
Procedure for the estimation of the correct power dissipated by the braking rheostat.

**Figure 13 sensors-20-06935-f013:**
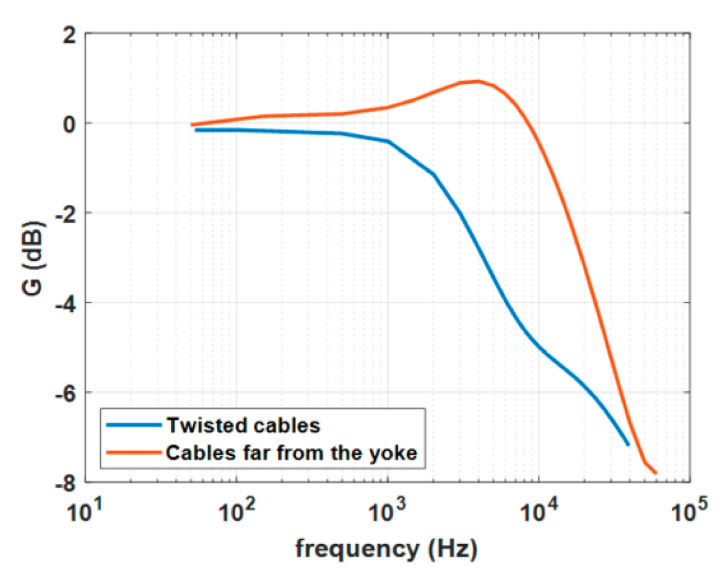
Frequency behavior of the current sensor gain for two different arrangements of the primary conductor.

**Figure 14 sensors-20-06935-f014:**
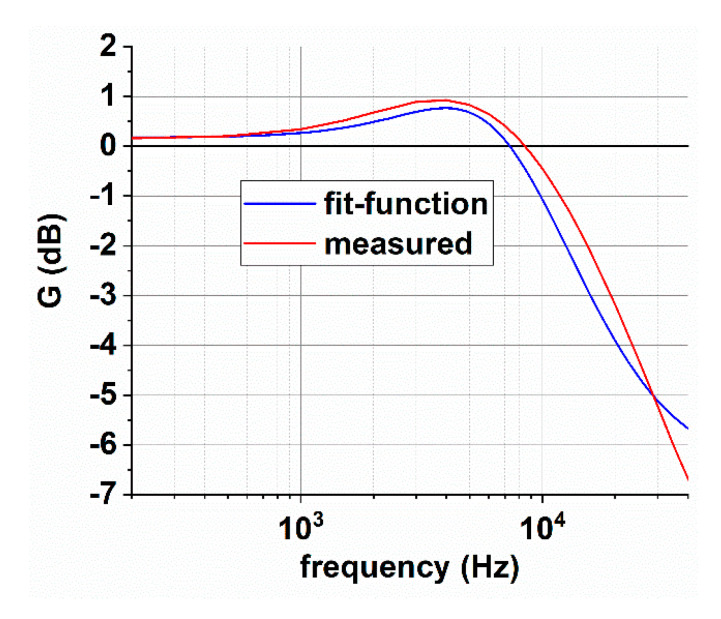
Comparison between frequency function of the identified fit-function and the measured one.

**Figure 15 sensors-20-06935-f015:**
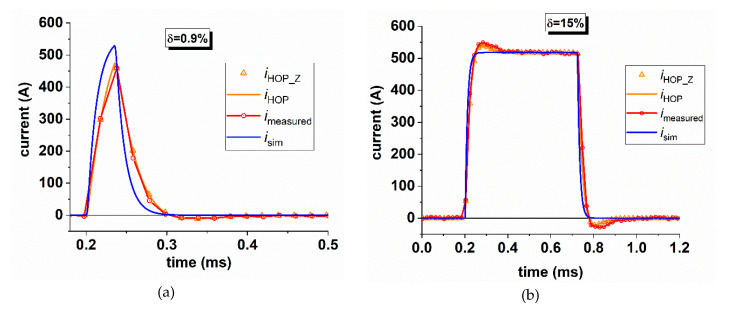
Comparison between the actual current flowing in the rheostat, the distorted signal provided by the current sensor and the digitization effects for a duty-cycle of 0.9% (**a**) and 15% (**b**).

**Figure 16 sensors-20-06935-f016:**
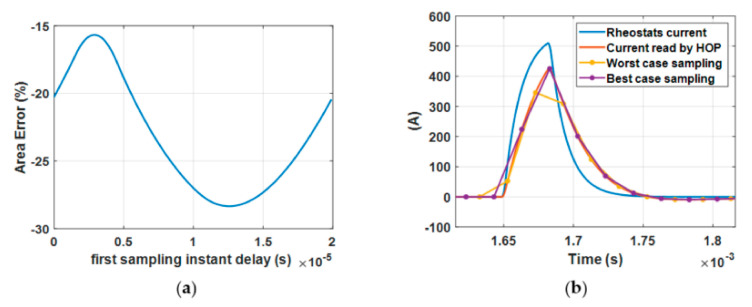
Impact of the position of the first sample following the current pulse for a duty-cycle of δ = 0.9% (**a**). A comparison between signals obtained by sampling the current information provided by the current transducer with a different position of the first sample (**b**).

**Figure 17 sensors-20-06935-f017:**
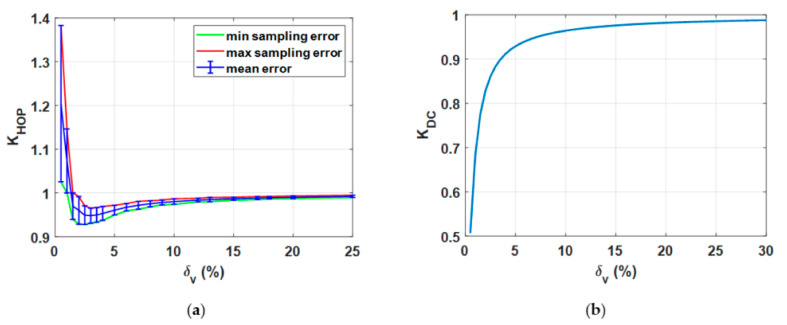
Correction coefficient *K*_HOP_ versus the chopper duty-cycle *δ*_V_ (**a**). Correction coefficient *K*_DC_ versus the chopper duty-cycle *δ*_V_ (**b**).

**Figure 18 sensors-20-06935-f018:**
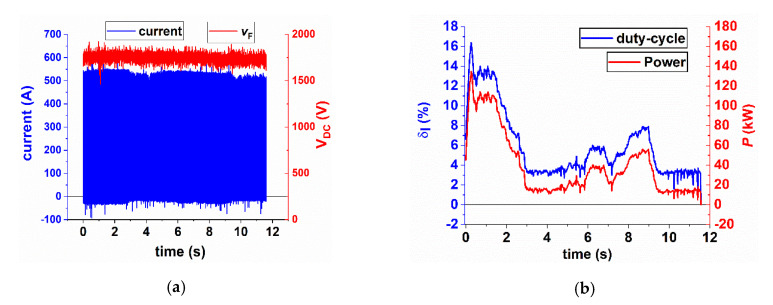
Dissipative electric braking taken as test for the proposed correction procedure (**a**). Time behavior of the dissipated power and the current duty-cycle (**b**).

**Figure 19 sensors-20-06935-f019:**
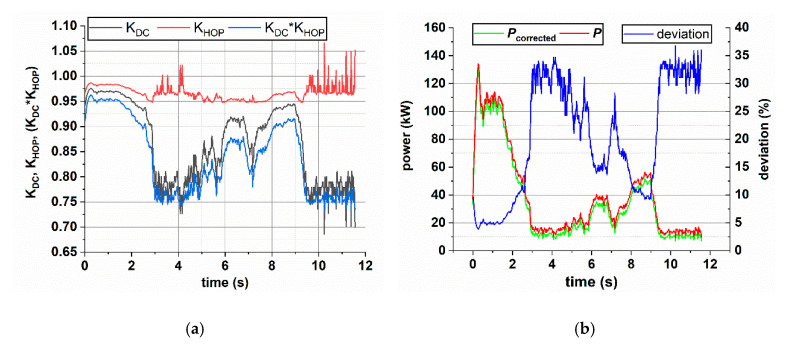
Time behavior of the correction coefficients (**a**), the power *P*, the corrected power *P*_corr_, and their deviation (**b**).

**Table 1 sensors-20-06935-t001:** Correction factor *K*_DC_ for different chopper duty-cycle *δ*.

*δ* (%)	*K* _DC_
0.7	0.633
2	0.850
4	0.923
8	0.961
10	0.969
15	0.979
30	0.990

**Table 2 sensors-20-06935-t002:** Summary of the main specifications of the LEM HOP 800 current sensor.

Primary nominal RMS current	800 A
Output voltage	± 4 V
Supply voltage (± 5%)	12 … 15 V
Accuracy	≤ ± 2%
Slew rate	50 A/µs
Voltage insulation	2000 V
Frequency bandwidth (−1 dB)	DC … 10 kHz

**Table 3 sensors-20-06935-t003:** Correction coefficients *K*_HOP_ and *K*_DC_, and their product with the associated interval of maximum variation due to the sampling effect.

*δ* _v_	*K* _HOP_	*K* _DC_	(*K*_DC_∙*K*_HOP_)
0.7	1.124	0.633	0.711 ± 0.064
2	0.960	0.850	0.816 ± 0.027
8	0.975	0.961	0.937 ± 0.007
10	0.980	0.969	0.950 ± 0.006
15	0.987	0.979	0.966 ± 0.003
30	0.993	0.990	0.983 ± 0.002

**Table 4 sensors-20-06935-t004:** Comparison between the power measured *P* and the corrected power for different values of the current duty-cycle.

*δ* _I_	*P* (kW)	*P*_corr_ (kW)	Δ (%)	± Var (%)
3	13.8	10.3	34	4.3
7.4	54	49	10.2	1.0
10.6	84.2	79	6.6	0.65
16.4	134	129	3.9	0.37

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
