# Peer review of "Methodology for the Accurate Measurement of the Power Dissipated by Braking Rheostats"

_sensors, 2020, doi:10.3390/s20236935_

Round 1
Reviewer 1 Report
The authors have revised the paper and addressed my concerns.
Author Response
The authors would like to thank the reviewer for her/his contribution
Reviewer 2 Report
1) With the term "period" I refer to the duration.
For instance, should the user collect data that cover a period
of 1 min, 1 h, etc. prior to the analysis?
2) What quantities are measured (i.e. currents, voltages, etc.)?
3) How many metering units are needed (i.e. how many meters to measure currents, etc.)?
4) This comment is related to 1)
How much data are necessary to verify the results and draw conclusions?
5) What happens in cases when data are missing due to metering failures?
For instance, we want to collect 10 current measures in sequence and entries 3 and 4 are missing.
6) How extreme values are detected?
For instance, how can we conclude that an extreme value of a measured current is common and not
an expected extreme value (outlier)?
Author Response
Please see the attachment

This manuscript is a resubmission of an earlier submission. The following is a list of the peer review reports and author responses from that submission.
Round 1
Reviewer 1 Report
The paper presents a method for compensating systematic effects affecting power measurement in rheostatic braking systems. The paper is quite well written and its technical content is scientifically sounding; however, the presentation quality has to be hardly improved. The key operating step of methods are suitably described, even though few explanations are still required. Results obtained in both laboratory and actual tests confirm the promising performance of the authors’ proposal.
Few concerns have to be addressed before the final paper publication
- The paper has to be accurately checked for English usage by a mother-tongue person; several typos are also present throughout the text. Some examples are
- 92 “a maximum. 50 %.”
- 151 “It apparent that this approach”
- 186 “transducer is use in condition much different of those of”
- 190 “is -1 dbB.”
- 284 “frequency behavior”
- 286 “of the function that fit with acceptable”
- Legend of Fig.13 “fit-fuction”
- 303 “3.3.Sempling effects in the”
- 376 “KHOP which time”
- Please, provide the measurement unit for HOP accuracy in Tab.II
- Please, define always acronyms, even though they appears trivial (e.g. ADC at line 216)
- What does the subscript “s” mean in the symbol is (l. 218)
- In the evaluation of KHOP, it's not clear how the authors take into account the frequency limitations in their lab tests.
- The sentence “The frequency band is considerably lower than the rated one, about -1 dB at 2 kHz.” at line 296 refers to twisted cable configuration?
- What is the effect of the 10% variability of KHOP for delta_v equal to 0.7%on the final estimate of power dissipated in the rheostat?
- Vertical labels of the second axes system in Fig. 17.b are cut
Reviewer 2 Report
The work is interesting. The reviewer has the following comments:
Author affiliations are not complete.
“The optimized recovery of the energy provided by the electrical braking can lead to saving from about 10% to 30%.” It is not clear if this is identified from previous work or other discovery. How is this value linked to the statement “Using the proposed technique, the estimation of the dissipated braking energy is improved by 20%.”?
“Tools for the accurate estimation of the amount of the potential energy that can be saved allow a more reliable cost-benefit assessment.” It is not clear where the cost-benefit assessment is in this work and the implication is not discussed.
Keywords are not provided.
The literature review is inadequate. For example, on page 2 lines 58-59, the authors state that “An estimation technique that exploits the electrical measurement on the braking rheostat has been briefly described in [7].” It is not clear if this work is additional work to this reference or not. Also, what is the gap in knowledge of [7]?
The structure of the paper is not clear and the introduction is poorly written. After the introduction, the authors should state what the other sections will present. In addition, the gap in knowledge has not been identified.
Figure 2, please adopt the IEC standard for the components.
Symbols need to be defined in nomenclature. Also, a serious issue is the symbols defined in the text are different from the symbols used in the equations, e.g. Equation 1 for IR and VR.
Figure 10 should have the figure caption with the figure.
Figure 17, the red axis is missing labels.
Correct English, “is of About 20%.”
The conclusion is missing.